# Small Leucine-Rich Proteoglycans (SLRPs) in the Retina

**DOI:** 10.3390/ijms22147293

**Published:** 2021-07-07

**Authors:** Shermaine W. Y. Low, Thomas B. Connor, Iris S. Kassem, Deborah M. Costakos, Shyam S. Chaurasia

**Affiliations:** Ocular Immunology and Angiogenesis Lab, Department of Ophthalmology and Visual Sciences, Medical College of Wisconsin, Milwaukee, WI 53226, USA; wlow@mcw.edu (S.W.Y.L.); tconnor@mcw.edu (T.B.C.); ikassem@mcw.edu (I.S.K.); dcostakos@mcw.edu (D.M.C.)

**Keywords:** retina, small leucine rich proteoglycans (SLRP), biglycan, decorin, fibromodulin, lumican, PRELP, opticin, osteoglycin/mimecan, chondroadherin, tsukushi, nyctalopin

## Abstract

Retinal diseases such as age-related macular degeneration (AMD), retinopathy of prematurity (ROP), and diabetic retinopathy (DR) are the leading causes of visual impairment worldwide. There is a critical need to understand the structural and cellular components that play a vital role in the pathophysiology of retinal diseases. One potential component is the family of structural proteins called small leucine-rich proteoglycans (SLRPs). SLRPs are crucial in many fundamental biological processes involved in the maintenance of retinal homeostasis. They are present within the extracellular matrix (ECM) of connective and vascular tissues and contribute to tissue organization and modulation of cell growth. They play a vital role in cell–matrix interactions in many upstream signaling pathways involved in fibrillogenesis and angiogenesis. In this comprehensive review, we describe the expression patterns and function of SLRPs in the retina, including Biglycan and Decorin from class I; Fibromodulin, Lumican, and a Proline/arginine-rich end leucine-rich repeat protein (PRELP) from class II; Opticin and Osteoglycin/Mimecan from class III; and Chondroadherin (CHAD), Tsukushi and Nyctalopin from class IV.

## 1. Introduction

Retinal diseases are one of the leading causes of visual impairment worldwide. Amongst working adults and the elderly, an estimated 8.7% of adults aged 45–85 years are affected by age-related macular degeneration (AMD) [1], and 93 million people are suffering from diabetic retinopathy (DR) [2] globally. In infants, a proliferative retinal vascular disease commonly described as retinopathy of prematurity (ROP) is the most common cause of childhood blindness that affects 14,000–16,000 babies born in the United States annually [3]. This is concerning, especially with increasing incidences of ROP worldwide and a third epidemic presenting multiple challenges to low- and middle- income countries [4,5]. With retinal diseases on the rise, there is an unmet need in understanding the different structural and cellular components that play a critical role in the pathophysiology of these diseases.

One potential structural component that has played a substantial role in several ocular diseases includes small leucine-rich proteoglycans (SLRPs). They are present within the extracellular matrix (ECM) of connective and vascular tissue where they mediate various cell–matrix interactions [6,7]. Abnormal polymorphism or expression of these SLRPs may lead to abnormal ECM or tissue physiology and result in various pathological conditions, including fibrosis, inflammation, angiogenesis, and even cancer.

In the eye, many inherited SLRP-linked genetic diseases are known to cause several ocular abnormalities [8]. Few studies have suggested that proteoglycans determine axonal guidance from the retina [9] and help maintain adhesion between the retinal pigmented epithelial (RPE) cells and the neurosensory retina [10]. Associations between several SLRPs and high myopia have also been suggested [11,12]. Furthermore, SLRPs have been shown to play a role as pro- and anti-angiogenic factors that can modulate retinal vasculature. Additionally, they are involved in many upstream signaling pathways such as the receptor tyrosine kinase, Toll-like receptor and BMP/TGF-β receptor pathways [8] that are involved during retinal development [13,14].

SLRPs consists of two main components, a conserved core protein and variable glycosaminoglycan (GAG) side chains. The core proteins comprise of leucine-rich repeats (LRRs), units of around 20–29 amino acids with a hallmark consensus sequence of LXXLxLXXNxL. L, leucine can be replaced by isoleucine, valine, and other hydrophobic amino acids, whilst x can be any amino acid [15]. These LRRs are flanked by a sequence of four cysteine residues at the *n*-terminus and two cysteine residues at the C-terminus [16]. The core proteins take on a secondary structure constituting an α-helix, short parallel β-sheets, and a more variable region. The side chains protrude from the short parallel β-sheets and mediate protein-specific interactions [17]. The GAGs, on the other hand, are linear polysaccharides that are negatively charged and can be either sulphated or non-sulphated.

In light of the multiple functional roles that SLRPs play in regulating biological processes, there has been enormous interest in elucidating their roles in retinal health and disease. This comprehensive review thus serves as a platform to summarize our current knowledge on SLRPs in the retina and provide future directions to understand their role in the pathophysiology of retinal diseases.

## 2. Classification of Retinal SLRPs

SLRPs can be broadly divided into five distinct classes based on their number of LRRs, amino acid residues at the N-terminus and their chromosomal organization. In canonical classes I–III, a capping motif, comprising of two terminal LRR and an “ear repeat”, can be found. These ear repeats maintain the protein core’s structural conformation and influences its ligand binding ability [18]. On the other hand, Class IV and V SLRPs do not have ear repeats [16]. Figure 1 summarizes the gene–protein information for each SLRP that has been found in the retina. Table 1 summarizes the distribution of known SLRPs in the retina.

## 3. Class I Retinal SLRPs

Class I SLRPs encompass a cluster of C_x_3C_x_C_x_6C N-terminal cysteine residues with two disulfide bonds. They have highly conserved intron and exon junctions and share a similar organization of eight exons [8]. Their GAG sidechains are typically chondroitin sulfate and dermatan sulfate consisting of repeated acetylated amino sugar moieties including N-acetyl-galactosamine or -glucosamine, and D-glucoronic or L-iduronic acid [32].

### 3.1. Biglycan

Biglycan is a member of the class I SLRP family [8]. Its 14 kB gene is mapped to the human Xq28 chromosome and encodes for a core protein of 42 kDa comprising 12 LRRs [33]. Biglycan was described as a neurotrophic factor in retinal cells and may play a role in regulating cell differentiation, angiogenesis, and retinal structural integrity.

In humans, biglycan has been found in all the layers of neurosensory retina. Immunohistochemistry detected its presence in the Bruch’s membrane, choroid and sclera of the retina [19]. It was also localized in choroidal vessels and retinal blood cells, including leukocytes and erythrocytes [19]. Biglycan distribution has also been investigated in mice during both adult and developmental stages. It was found to be present in the retina and optic nerve by embryonic day 18 (E18) with high densities in the laminar beam located at the lamina cribrosa [34]. On postnatal day 7 (P7), biglycan was detected in the optic nerve, ganglion cell layer (GCL) and inner nuclear layer (INL). It was also diffusely distributed in the outer retina. By P42, biglycan displayed intense staining in retinal layers rich with synapses such as the nerve fiber layer (NFL), inner plexiform layer (IPL) and outer plexiform layer (OPL).

The regulation of biglycan expression may be crucial for overall retinal health. Binding of biglycan to apoB-100 has played a central role in plasma lipoprotein entrapment in artery walls [35]. This disrupts the transportation and exchange of nutrients and metabolic products. Disruptions across the choroidal blood vessels and the Bruch’s membrane can contribute to photoreceptor dysfunction, atrophy, neovascularization and RPE detachment. Mice overexpressing the apolipoprotein B-100 (apoB-100) gene with biglycan is more susceptible to hyperlipidemia, Bruch’s membrane thickening and sub-RPE particle formation when fed a high cholesterol diet [36]. Therefore, overexpression of biglycan can potentially induce complications associated with retinal integrity and may result in devastating retinal diseases. A correlation between biglycan gene expression and an increase in Bruch’s membrane thickness has also been observed [36], indicating that the regulation of biglycan may be crucial in maintaining retinal health.

Biglycan may also play a role in retinal microvasculature. Although still uncertain, a study has shown that bovine pericytes and endothelial cells can contribute to biglycan synthesis and distribution [37]. When exposed to fibroblast growth factor, an increase in biglycan expression was observed in endothelial cells. Biglycan may hence be involved in regulating endothelial cell proliferation and angiogenesis. Additionally, biglycan has also been found to be highly associated with angiogenic markers such as vascular endothelial growth factor (VEGF), platelet derived growth factor (PDGF), and angiopoietin like proteins (ANGPTLs) in an in silico gene analysis of gastric cell tissues [38]. Moreover, biglycan deficient cells formed a smaller number of new vessels in chick embryo chorioallantoic membrane xenografts. Migrating cells during scratch wound assay showed elevated expression of biglycan [39]. Nevertheless, studies related to biglycan’s role in retinal vascularization are meagre.

Besides its involvement in regulating retinal health, biglycan has been suggested as phototoxicity biomarkers related to UV-induced damage to the interphotoreceptor matrix (IPM). UVA and UVB irradiation to rat retinas have been shown to result in the loss of biglycan expression [30]. This loss in biglycan expression due to UV exposure may lead to structural changes in the ECM and damage to the retina.

Discordant to previous findings, a study investigating kainic acid-induced retinal detachment found weak biglycan staining patterns for both normal and detached retinas, suggesting that biglycan may have a limited or insignificant role in retinal repair. Biglycan mRNA expression was also reported to be low in both control and experimental groups [20]. A possible explanation for this could be the compensatory activation of the closely related SLRPs in the biglycan deficient mice [40]. This reasoning may be plausible considering decorin, a similar SLRP, was more strongly distributed in the kainic acid mouse retinas [20].

### 3.2. Decorin

Decorin is another member of the class I SLRP family [8]. The decorin gene is approximately 42 kB in size and is mapped to human chromosome 12q21.33. Its core protein comprises 12 LRRs and is approximately 39 kDa in size. Decorin is involved in various biological functions, including the modulation of ECM, growth factors, cell proliferation, survival, migration, and angiogenesis [41].

Decorin can be found in all layers of the human retina [19]. Decorin has also been demonstrated in rat retinal tissue [21]. Immunostaining for decorin core protein was present throughout the entire rat retinal tissue during its development. By P14, strong staining densities were observed in the NFL and GCL, moderate densities were observed in the IPL, INL, and RPE and faint densities were observed in the outer nuclear layer (ONL) [21]. Similarly, in mice, decorin is diffusely distributed in embryonic and early postnatal mice. By P42, decorin was strongly detected in the NFL, GCL and photoreceptor layer (PRL) [20]. It was more diffusely distributed in the IPL, INL, OPL, and ONL.

Decorin has been reported as a neurotrophic factor and is known to play a crucial role in retinal cell differentiation, growth, repair, and survival. Decrease in decorin levels can reflect cellular damage and cell death whilst the recovery of decorin can occur in response to retinal regenerative processes or be associated with several inflammatory responses. Oxygen induced retinopathy (OIR) in Sprague–Dawley rats displayed significant decrease in decorin expression as compared to their controls [42]. Another study also found similar results whereby OIR rats with severe retinal injury and neuronal cell death presented significantly reduced decorin mRNA and protein expression [43]. Interestingly, treatment with triamcinolone acetonide (TA) after the hyperoxic phase was able to prevent the decrease in decorin mRNA and protein expression, improve immunoreactivity levels in the GCL and INL and was able to restore neuronal cells [43]. Immunofluorescent staining also exhibited an overlap between the decorin and neuronal nuclei signals, suggesting that decorin plays a role in neuronal regeneration. In another study, the effects of decorin have been compared to the anti-VEGF bevacizumab in OIR rats [44]. OIR rats that were treated with 0.1 mg/kg of decorin presented decreased VEGF and TNFα immunoreactivity comparable to bevacizumab. This exciting study highlights the beneficial use of decorin for the treatment of hypoxia related vascularization commonly associated with retinal diseases such as ROP.

Looking into the distribution of decorin in a mouse model with damaged retina, mice were injected with kainic acid to induce progressive inner retinal cell loss [20]. Three days post-injection, decorin levels were upregulated with its distribution localized to the inner retinal layers and diffused throughout the outer retinal layers. In the later stages when there was significant loss in inner retinal cells, decorin was present throughout the entire mouse retina. Similarly, in a study by Inatani in 1999, an ischemic reperfusion rat model presented a transient downregulation of the decorin core protein gene in the GCL and INL after 24 to 48 h, but during the later stages, presented strong presence of decorin in the damaged inner retina, with co-localization to the retinal ganglion cells (RGCs) [21]. This suggested that decorin may be involved in the differentiation of ganglion cells and the survival of retinal cells.

Like biglycans, decorin has also been shown to be synthesized by bovine retinal pericytes [37]. Decorin is unique as it displays both pro-angiogenic and anti-angiogenic effects [45]. CNV is associated with wet AMD, which often result in severe vision loss when associated with hemorrhage. Decorin may play a key regulator for angiogenic responses by acting as an inhibitor for multiple receptor tyrosine kinases (RTKs) such as the epidermal growth factor receptor (EGFR) [46], the insulin-like growth factor receptor I (IGFR-1), [47] and the Met hepatocyte growth factor receptor (HGFR) [48]. It has also been shown to decrease hypoxia-induced vascular endothelial growth factor expression via blocking the Met expression pathway and downregulating Ras-related C3 botulinum toxin substrate 1 (Rac1) and Hypoxia-inducible factor 1-alpha (HIF-1α) [49]. Another study has also shown decorin to be a VEGF-R2 antagonistic ligand and suggested that decorin may suppress hypoxia induced-VEGF expression and angiogenesis in RPE cells [50].

Decorin could potentially be used in therapeutic treatment to prevent or regress neovascularization that often occurs in retinal angiogenic diseases. For example, the overexpression of decorin via lentivirus vector significantly prevented the proliferation, migration, and tube formation in hypoxic ARPE-19 cells [49]. Additionally, intravitreal treatment of decorin significantly suppressed VEGF and TNF-α expression and inhibited TGF-β in the RPE-choroid complex in a mouse model of laser-induced CNV [51]. Mice pretreated with decorin presented reduced fluorescein vascular leakage and Isolectin B4 staining in areas of CNV formation. This demonstrated that the administration of decorin can inhibit laser-induced CNV both before and after the onset of angiogenesis. Decorin can also potentially be used in PVR treatment strategies. In human RPE cells, TGF-β2 induces epithelial mesenchymal transition (EMT) and encourages epithelial cells to differentiate into matrix producing fibroblasts and myofibroblasts that contribute to PVR [52]. Decorin, being a TGF-β antagonist, may thus have the potential to suppress the progression of EMT and therapeutically prevent PVR.

Decorin can also modulate matrix metalloproteinase (MMP) activity by altering plasminogen concentrations to favor ECM degradation [53]. In vitro, decorin has been found to inhibit EMT and fibrosis by reducing fibronectin, laminin, vimentin, collagen I, and α-SMA expression in ARPE-19 cells [54]. However, despite its involvement in modulating TGF-β and EMT, it is unlikely that decorin levels will be useful as predictors for PVR [54]. In rabbits-induced with traumatic PVR, endogenous decorin was unable to reduce fibrogenic growth factors. However, the introduction of 100 or 200 µg of decorin was able to significantly reduce fibrosis development [55]. When tested in human subjects, a single intravitreal injection of 200 or 400 µg of decorin 48 h after PVR injury has shown no adverse effects and patients depicted no significant worsening of PVR. Decorin levels were reported to be significantly elevated by the fifth day post-injection and concentrations were maintained high until the eighth day [56].

Besides regulating EMT in PVR, decorin has also shown encouraging results in preventing glaucomatous retinal injuries. In a rat model of glaucoma injected with TGF-β injections, treatment with human recombinant decorin (hrDecorin) reduced trabecular meshwork (TM) fibrosis, increased MMP2 and MMP9 levels, lowered TIMP2 levels, reduced intraocular pressure (IOP) and prevented progressive RGC loss. However, in vitro, treatment of retinal cells with hrDecorin did not significantly change βIII tubulin RGC frequency, suggesting that decorin does not have direct neuroprotective effects and RGC protection in vivo may be attributed to decorin mediated IOP reduction [57].

Chronic hyperglycemia plays a key role in the pathogenesis of DR. Human ARPE-19 cells that were cultured in high glucose and treated with 100 nM of decorin resulted in significant reduction in apical–basolateral permeability, increased transepithelial electrical resistance (TEER), elevated tight junction proteins such as occludin and zonula occludens-1 (ZO-1) and reduced phosphorylation of p38 MAPK [58]. Decorin thus has the potential to be a useful therapeutic for DR.

Figure 2 summarizes decorin’s versatile role in modulating retinal homeostasis during damage, hypoxia, increased barrier permeability, fibrillogenesis, and angiogenesis.

## 4. Class II Retinal SLRPs

Class II SLRPs are polyanionic and contain tyrosine sulfate residue clusters at their N-terminus with a cys-rich cluster consensus of C_X3_C_X_C_X9_C. Their GAG members are primarily keratan sulfate and polylactosamine. However, in the human retina and choroid, core proteins have been reported to contain keratan sulfate [59]. The exons of class II SLRPs generally share a similar organization, with a large central exon that encodes most of their 12 LRRs [8].

### 4.1. Fibromodulin

Fibromodulin is a member of the class II SLRP family [16]. The core protein is approximately 43 kDa in size and has a core protein that is similar to decorin and biglycan. The human fibromodulin gene is around 11.1 kB and is located at the 1q32.1 chromosome [33]. It is known to interact with type I and II collagen fibrils [60] and plays a role in inhibiting fibrillogenesis.

In the human eye, fibromodulin can be found strongly distributed in all layers of the retina and moderately distributed in the Bruch’s membrane, choroid and sclera [19]. In prenatal and early postnatal mice, fibromodulin can be found weakly distributed in the retina. By adulthood, they are more moderately distributed with high affinity to the NFL, GCL, IPL, and INL [20].

Fibromodulin might contribute to the damage and repair processes in the retina. In mouse eyes injected with kainic acid to induce progressive loss of inner retinal cells, fibromodulin was detected to be strongly present in the inner retinal layers [20]. During the later stages when there was significant loss of inner retinal cells, fibromodulin could be detected throughout the entire retina including the PRL. Additionally, fibromodulin mRNA expression was found to be upregulated after kainic acid injection. Another study also found that fibromodulin deficient mice presented with multiple areas of retinal detachment, suggesting that fibromodulin may be vital in the maintenance of retinal adhesion [11]. However, these fibromodulin deficient mice do not present significant RGC axon loss when induced with glaucoma [61]. This suggests that fibromodulin may not be involved in the loss of RGC observed in glaucomatous eyes.

One potential mechanism by which fibromodulin may modulate retinal health is through the complement pathway. Fibromodulin has been shown to regulate the complement pathway by binding to complement inhibitor factor H [62]. Interfering with fibromodulin associated complement pathway may thus serve to be beneficial in retinal diseases. Recently, a study has found that the serine protease high temperature requirement A1 (HTRA1) promoter variant is a strong genetic locus that is linked with AMD [63]. The HTRA1 protein, expressed by the RPE, was able to cleave fibromodulin with 90% efficiency after 20 h of treatment. Preventing the breakdown of fibromodulin may thus be beneficial in maintaining retinal health and integrity.

### 4.2. Lumican

Lumican is another member of the class II SLRP family [16]. The human lumican gene is located at the 12q21.33 chromosome. It is involved in collagen fibrillogenesis regulation, the modulation of fibril diameter spacings and the regulation of proinflammatory responses.

In the eye, lumican can be found in the human neurosensory retina, retinal blood cells, the Bruch’s membrane, the choroid and the sclera [19]. Lumican expression levels are closely associated with light exposure. Lumican transcripts in neonatal mouse retina have been found to increase progressively in response to light stimuli. Its levels have also been reported to have increased more than 150 times in three days after light exposure, and was found to be located at the GCL, IPL, INL, and OPL of the mouse retina [22]. Interestingly, the accumulation of lumican was correlated with RGC loss from increased exposure to light. However, it is not clear that lumican acts as a regulatory factor which may cause abnormalities or loss in RGC. The increased levels of lumican are a part of two different mechanisms. Lumican has also been identified as a UV phototoxicity biomarker associated with interphotoreceptor matrix (IPM) disruption [30]. Rat retina that was exposed to UVB had significantly reduced lumican expression whilst retina treated with UVA resulted in downregulation of lumican expression. The reduced expression of lumican has been found to be associated with impaired structural integrity and function of the IPM. Lumican is also involved in EMT. Smad3, a mediator of TGF-β, has been shown to be vital for the expression of lumican in the eye [64]. In Smad3 knockout mice, lumican and EMT was absent post-retinal detachment, indicating the role of Smad3 in lumican expression. Considering lumican’s close association with EMT activation, its expression has been identified as one of the hallmarks for EMT [65].

Lumican may also be closely associated with fibromodulin. Lumican and fibromodulin double knockout mice have shown several characteristics of high myopia, including increased axial length (10%) and retinal detachment (80%) [11]. Double knockout eyes also presented significantly more subretinal debris (>50%). Alterations to these SLRPs may thus contribute to myopia and various retinal diseases.

### 4.3. Proline/Arginine-Rich End Leucine-Rich Repeat Protein (PRELP)/Prolargin

Proline/arginine-rich end leucine-rich repeat protein (PRELP) or prolargin is another proteoglycan of the class II SLRP family [16]. Its gene has been mapped to the human 1q32.1 chromosome and its 44 kDa core protein aids in anchoring collagen to basement membranes or to its underlying connective tissues such as cartilage. The N-terminus of PRELP binds to both heparin and heparan sulfate proteoglycans in the ECM [66].

In the human eye, PRELP can be found distributed strongly throughout the retina and moderately through the Bruch’s membrane, choroid, and sclera [19]. When delivered to mice retina, PRELP was found to be localized in the RPE and outer segments of the retina with perinuclear staining in the ONL. It was suggested that the photoreceptors and RPE synthesize and secrete PRELP which later localize to the inner and outer retina [29].

PRELP may play a role in the prevention of CNV and AMD by inhibiting the complement pathway. The complement system has been known to converge into a terminal pathway, which results in the deposition of the membrane attack complex (MAC). PRELP has been recently identified to block a crucial process in the assembly of MAC by inhibiting C9 polymerization. It thereby protects against the formation of CNV and AMD. In C57Bl/6J mice injected with a vector carrying PRELP, a significant reduction in laser induced CNV (60%) and MAC deposition (25.5%) was observed [29]. However, in human umbilical vein endothelial cells (HUVECs), the introduction of PRELP presented an increase in the formation of master junctions, master segments, meshes and tubes in the media, suggesting that the suppression of tube formation in vivo was prevented by the complement cascade [29]. In another study, PRELP has also been shown to inhibit C3 formation, which is involved in the alternative pathway of complement system [67]. These studies demonstrate that PRELP may regulate different stages of the complement pathway. The development of PRELP therapeutics may hence be helpful against CNV.

## 5. Class III Retinal SLRPs

This class of SLRPs have a N-terminus with a cys-rich cluster consensus of C_X2_C_X_C_X6_C and a core that is generally made of 7 LRRs [8].

### 5.1. Opticin

Opticin is a member of the SLRP class III family which shares high LRR homology with epiphycan and osteoglycin [16]. Its core protein is around 37 kDa in size and has been found to be strongly associated with bovine collagen fibrils found in the vitreous humor [68]. Its gene is located in the 1q31-q32 region where at least three genetic retinal diseases, including AMD, can be found [69]. Opticin has sites for N- or O-linked oligosaccharide substitution and has no GAG side chains [70,71].

In the human eye, opticin has been identified in the cornea, iris, ciliary body, vitreous, retina, and choroid. From immunoblot analysis, opticin is approximately 38, 45, or 65 kDa in size after post-translational modifications [69,72]. In primary cultured human RPE cells, 60 kDa, 45 kDa, and 38 kDa bands were observed, with opticin localized to the cytoplasm [73]. Extraocular blood vessel adventitia, extraocular nerve epineurium, and intraluminal blood vessels were also labelled for opticin [72]. Immunohistochemical analysis of opticin revealed a gradient, with the strongest stains found in the internal limiting membrane (ILM) which slowly fades towards the choroid and optic nerve head [19]. It has been suggested that opticin could be secreted by the non-pigmented ciliary epithelium [74] before diffusing from the ciliary body to the vitreous body and retina. In the rat eyes, opticin has been identified as a 37 kDa or 50 kDa protein [69] whilst in the chick eyes, a 60–62 kDa or 45 kDa protein has been identified [13]. Opticin mRNA expression has also been reported in the retinal layers of canine [31] and in the ciliary body pigmented epithelium of developing mouse models [75]. The difference in opticin sizes between human and other species may be due to interspecies differences in post-translational modification.

Opticin’s localization to the ILM suggests its role in vitreoretinal adhesion. The vitreous gel has been demonstrated to have positive immunoreactivity with opticin, especially at gel membrane interfaces where individual vitreous fibrils condense [76]. Other studies have also shown opticin to be observed in the vitreous collagen and ILM [77]. Opticin may thus be involved with macromolecular surface coatings which can indirectly contribute to the spaces between vitreous collagen fibrils and adhesion with the ILM. Opticin may hence provide interesting insights in its role in PVD or aging, since this coating is more predisposed to disruption with age. This disruption may eventually lead to gel liquefaction and the weakening of vitreoretinal adhesion [78].

Opticin has also been demonstrated to have in vivo and in vitro anti-angiogenic properties. Administrating opticin to the sites of angiogenesis in mice significantly inhibited neovascularization. This inhibition was not dependent on the type of growth factor used to induce angiogenesis, suggesting that the anti-angiogenic activities of opticin are not growth factor specific. However, they are specific to the matrix and can decrease capillary morphogenesis in type I collagen and in Matrigel [79]. Opticin acts as a competitive inhibitor and prevents the adhesion of endothelial cells to collagen, thereby preventing pathological angiogenesis [79]. This is supported by another study, whereby opticin deficient OIR mice depicted significant increase in preretinal neovascularization [80]. On the contrary, neovascularization in wildtype mice was inhibited by intravitreal delivery of opticin, suggesting its role in preventing angiogenesis.

In human primary RPE cells, mRNA expression levels of opticin are not affected by hypoxia, MMP, or VEGF stimulation. However, they can alter the levels of extracellular opticin in the culture medium. Both hypoxia and MMP-2 treatments resulted in decreased levels of extracellular opticin in vitro [73]. This is in line with other studies where MMPs can degrade opticin [81,82]. However, opticin degradation can be reversed with the introduction of an MMP inhibitor, ethylenediaminetetraacetic acid (EDTA). In vitreous samples, exogenous VEGF has also been reported to increase MMP-2 levels and decrease secreted opticin levels in a dose-dependent manner [73]. Furthermore, incubation with MMP-2 almost completely degraded opticin within 24 h, suggesting that opticin may be a target for MMP-2 proteolysis. Thus, neovascularization in retinal diseases such as PVR may be affected by opticin metabolism that is regulated by MMP-2.

Opticin may also serve as a potential biomarker for neovascular AMD. In a study analyzing 108 neovascular AMD patients, vitreous humor opticin levels in patients were observed to be reduced, suggesting that opticin may be involved in neovascularization [83]. However, additional information regarding the association of opticin and neovascular AMD is warranted. Opticin has also been shown to bind and regulate growth factors. In embryonic day 8 chicks, it was suggested that growth hormone (GH) is secreted by the RGCs and is sequestered and concentrated in the vitreous [13]. Opticin acts as an ocular GH binding protein and may serve to modulate GH activity by protecting it from degradation or by facilitating its extracellular transport. It may thus also be involved in retinal development by modulating ocular GH.

### 5.2. Osteoglycin/Mimecan

Osteoglycin or mimecan is another member of the class III SLRP family [16]. Its gene is mapped to the human 9q22.31 chromosome. In human eyes, osteoglycin was reported to be weakly present throughout the layers of the neurosensory retina but more strongly present in the Bruch’s membrane, choroid and the sclera [19]. Osteoglycin mRNA has also been detected in human cornea, iris, lens, and retina [84].

Few studies have been conducted on retinal osteoglycin. However, an osteoglycin interacting protein, leucine rich B7, has been found to be expressed in the human cornea, iris, retina and sclera [85]. B7 has two isoforms-isoform 1 detected in low levels in the retina whereas higher levels of isoform 2 were found in the retina. B7 protein expression has been reported to correlate with the sites of osteoglycin expression in the human PRL, however, little is known about its role in modulating retinal integrity. Further analysis of retinal osteoglycins and their interacting proteins can potentially aid us in better understanding its role in retinal health and physiology.

## 6. Class IV Retinal SLRPs

Chondroadherin, tsukushi, and nyctalopin are members of the class IV SLRP family [12]. This family has no ear repeats and have a core protein that is made of 11 LRRs. The N-terminal cys-rich region that carries a consensus sequence of C_X_3C_X_C_X_6-17C flanks its core protein.

### 6.1. Chondroadherin (CHAD)

Chondroadherin (CHAD) is a member of the class IV SLRP family [16] that has its gene mapped to the human 17q21.33 chromosome. It is a matrix protein commonly found in the cartilage that mediates chondrocyte adhesion.

CHAD has been detected in human cornea, lens, and retina [19] as well as demonstrated during retinal development [26]. In the mouse retina, CHAD immunolabelling was more prominent in the IPL, OPL, and PRL, suggesting its involvement in the maintenance of normal retinal physiology [26]. Its role may be similar to other SLRPs; however, not many studies have looked into CHAD expression in the retina. Further evaluation of this SLRP is required to address its functional roles in the retina.

### 6.2. Tsukushi

Tsukushi is a member of the class IV SLRP family [16]. Its gene is mapped to the human 11q13.5 chromosome and its expression pattern was what gave rise to its name; after the shape of the Japanese horse tail plant tsukushi. Tsukushi has been found to regulate multiple pathways in the extracellular signaling network. In adult mice, tsukushi is expressed in the lens epithelium, ciliary body, and the INL [28]. Its expression on the ciliary body suggests that it may play biological roles in the peripheral edge of the retina. Its expression in the INL also suggest that tsukushi may be involved in the regulation and proliferation of the Müller glia. Like CHAD, few studies have looked into the expression of Tsukushi in the retina. Further studies are needed to elucidate its functional role in the retina.

### 6.3. Nyctalopin

Nyctalopin is another member of the class IV SLRP family [16]. Its 28 kB human gene, Nyx, has been mapped to the human Xp11.4 chromosome. It is composed of three exons and it encodes a 481 amino acid protein which consists of a 13 LRR core protein. Nyctalopin is the first glycosylphosphatidylinositol (GPI) anchored member of the SLRP group. Disruption to this GPI anchor may result in abnormal retinal ON bipolar cell (BC) interconnections, which may result in the loss of nocturnal vision [8]. Nyx gene mutations have been associated with ocular diseases such X-linked congenital stationary night blindness (CSNB), nystagmus, and myopia.

Structurally, nyctalopin is attached to human cell membranes via a GPI anchor. It is likely that chimp, canine, and frog nyctalopin are also GPI anchored [86]. In murine, the LRR domain of nyctalopin has been found to be protruding out into the extracellular space whilst its core is anchored to the plasma membrane in a single transmembrane domain [87]. However, since mutations in the Nyx gene is distributed mainly throughout the LRR core sequence rather than the sequence of the GPI anchor, the mechanism of anchoring may be less crucial compared to the nyctalopin core protein present on the membrane itself.

In humans, nyctalopin has been reported to be expressed in the GCL, INL and PRL [23]. In the murine retina, nyctalopin mRNA and protein expression has been shown in the GCL, where it is expressed throughout all stages of postnatal retinal development and adulthood, and in the INL where it is associated with retinal amacrine cells [24]. The orthologous mouse Nyx gene was reported to have high degrees of conservation in its sequence and protein structure (83%) when compared to the human nyctalopin [24]. In terms of differences, its amino acid sequence at the N-terminus is five residues shorter than the human gene and its start codon coincides with the second start codon in the human sequence. In macaque and rabbit retina, nyctalopin has also been found to be located in the IPL, where it is associated with the BC terminals. In the OPL, it is associated with the photoreceptor and BC synapses [27]. Nyctalopin is also expressed in chick retina [25]. Its mRNA is expressed in the GCL and in the outer half of the INL of both developing and matured retina. However, chick nyctalopin only shares 55% similarity with the human nyctalopin. Zebrafish also express nyctalopin in the developing and adult retina at their ON BCs [88]. In adult zebrafish, nyctalopin expression was highest in the central retina and faded towards the periphery. Based on nyctalopin’s interactions and localization in the retina, it is likely that nyctalopin may be involved in the formation of the synapse or is involved in synaptic transmission of visual signals.

In the retina, rod and cone ON BCs signal via the metabotropic glutamate receptor 6 (mGluR6) pathway whilst OFF BCs signal via the α-amino-3-hydroxy-5-methyl-4-isoxazolepropionic acid (AMPA)/kainate receptor. A proteomic study revealed several nyctalopin-associated proteins in the mouse retina such as the transient receptor potential melastatin 1 (TRPM1) [89]. TRPM1 expression has also been reported to be dependent on nyctalopin in the OPL of the mouse retina [90]. TRPM1 was identified as a nyctalopin binding partner, with nyctalopin acting as an accessory for the localization of TRPM1 to the synapse [90]. mGluR6 has also been found to be closely associated with nyctalopin at the ON BC dendritic tips [27,91]. It was suggested that TRPM1 and mGluR6 form a macromolecular complex by physically associating with nyctalopin and that this assembly is vital in enabling fast signal transmissions for the ON BC response. Another protein, LRIT3 has also been reported to interact with nyctalopin and has been found to be crucial for nyctalopin expression in mice [92]. LRIT3 has structural and functional similarities to nyctalopin and has been shown to be required for the localization of nyctalopin to the ON BC dendritic tips. Loss of LRIT3 reduced the excitatory responses in both ON and OFF BCs and disrupted the normal functions of RGCs. In LRIT3 knockout mice, nyctalopin, and TRPM1 expression was also absent, indicating its role in maintaining healthy BC function. Ball and colleagues further elucidated nyctalopin’s role in rod BCs by analyzing the expression patterns of other synaptic proteins [93]. However, nyctalopin’s absence did not alter the expression patterns of many synaptic proteins including mGluR6, PKC, G0a, bassoon, PSD-95, a1F, trkB, and dystrophin. Rod BC dendritic morphology was also unaffected, suggesting that nyctalopin is not involved in the localization of these proteins to the BC. However, further evaluation of the functionality of these proteins are required to understand if nyctalopin is involved in BC signal transmission.

Nyx has been found to be mutated in patients with CSNB [23,94]. In the mutated Nyx, the segment required for creating the GPI anchor is eliminated and nyctalopin is unable to tether to cell membranes. In complete CSNB, the function of the rod system is completely absent, whilst the cone system is normal and can induce standard ERG measurements [95]. In a normal patient, scotopic 15 Hz flicker ERG will produce two signals, one for the rod BC pathway and another for the rod–cone coupling pathway. In patients with CSNB, complete blockage of the rod ON BC signal transmission can be observed, along with some residual signal transmissions through the rod–cone gap junctions of the OFF cone pathway [95]. However, in primate studies nyctalopin has been reported to act exclusively in the ON signaling pathway, with no evidence of involvement from the OFF pathway [96]. It is thus important to note that there are interspecies differences in nyctalopin’s involvement in rod and cone function. In a common model for human CSNB, nob mice with an 85 bp deletion in the Nyx gene presents reduced visual and light sensitivity and reduced electroretinography (ERG) b-wave amplitudes [97]. The 85 bp deletion results in a loss of 288 amino acids including 7 LRRs. This loss of function resulted in non-functional rod and cone ON BCs and reduced nyctalopin mRNA expression. Abnormal spontaneous activity in the RGC is also observed despite the lack of morphological abnormalities in the retina [91]. In nob rescue mice, nyctalopin was expressed in the early developmental stages and was able to restore both outer and inner visual functions [91,98]. However, in adult mice, nyctalopin expression was reduced by day 30 despite the relatively abundant levels of nyctalopin transcripts present, suggesting that there may be an obstacle in the post-translational process [98]. Therefore, supplementation of transgenes may only be effective when delivered in young differentiating BCs. Mutations in Nyx may also cause other ocular diseases such as myopia. Two unrelated male individuals have been reported to have novel mutations in the Nyx gene in distinct regions that are different from those related to CSNB associated myopia [12]. Nob mice have also displayed a significant myopic shift after two weeks of form deprivation whilst wildtype mice took six weeks to elicit a similar response [99]. Additionally, dopamine and 3,4-Dihydroxyphenylacetic acid (DOPAC) levels were also significantly lower in nob mice.

Nyctalopin has also been reported to regulate retinal activity required for the maintenance of axon terminal segregation in the dorsal–lateral geniculate nucleus between the two eyes [14]. In nob mice, eyes that had Nyx mutations failed to preserve precise eye specific territories. When rescued, transgenic mice had restored spontaneous retinal activity and the retinogeniculate circuitry was segregated and stabilized.

## 7. Other SLRPs Unknown to the Retina

Apart from the 10 SLRPs discussed above, there are seven other SLRPs yet to be studied in the retina, namely extracellular matrix 2 (ECM2) and asporin from class I; osteoadherin and keratocan from class II; epiphycan from class III; and podocan and podocan-like-protein-1 (Podnl1) from class V. Further evaluation of gene and protein expression to determine the presence and roles of these SLRPs in the retina can potentially expand new opportunities and possibilities for the understanding and treatment of devastating retinal diseases.

## 8. Conclusions

SLRPs can be found in abundance throughout the neurosensory retina. This article provides a comprehensive summary of the different retinal SLRPs and shares its current knowledge and impacts on retinal health and disease. These retinal SLRPs are involved in multiple signaling pathways and can modulate cell differentiation, adhesion, growth, and repair mechanisms. They interact with various molecules to exhibit anti-angiogenic effects, activate MMP and EMT, regulate the complement and are even involved in promoting synaptic signaling. Some retinal SLRPs have also been identified as effective biomarkers, which could potentially be targeted for intervention or used for diagnosis of retinal diseases. Additionally, several in vivo studies have been performed using SLPRs to mitigate the progression of retinal diseases such as CNV and PVD. Further research will direct SLRPs in translating their potential use in the clinic, especially with SLRPs that have been better established, such as decorin and nyctalopin. Studies looking into the less explored osteoglycin/mimecan, CHAD and tsukushi in the retina, and the seven other SLRPs which have yet to be evaluated in the retina, may also lead to promising new discoveries. With AMD, DR, and ROP on the rise, a deep understanding of these retinal SLRPs and their functional roles in the retina will be useful in the development of new therapeutic treatments against retinal diseases.

## Figures and Tables

**Figure 1 ijms-22-07293-f001:**
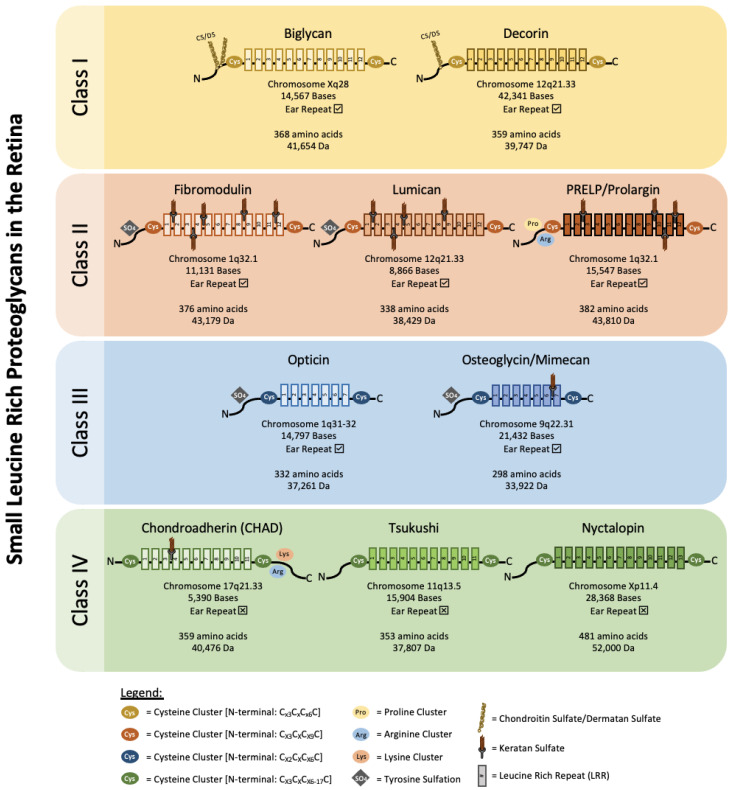
Classification and gene–protein information of known retinal SLRPs. Biglycan and decorin belong to class I SLRPs. They harbor chondroitin or dermatan sulfate side chains and have a protein core made of 12 LRRs. Their N-terminal cysteine clusters follow the sequence C_x3_C_x_C_x6_C. Fibromodulin, lumican, and PRELP/Prolargin belong to class II SLRPs. Class II SLRPs contain N-linked keratan sulfate and also have a protein core made of 12 LRRs. Their N-terminal cysteine cluster has the sequence C_X3_C_X_C_X9_C. Opticin and osteoglycin/mimecan are class III SLRPs. They both have a protein core made of 7 LRRs and a N-terminal cysteine cluster sequence of C_X2_C_X_C_X6_C. Osteoglycin/mimecan has a N-linked keratan sulfate side chain. Fibromodulin and lumican from class II and opticin and osteoglycin/mimecan from class III have clusters of tyrosine sulfate residues near their N-terminus. Chondroadherin (CHAD), tsukushi, and nyctalopin are class IV SLRPs. They have a protein core made of 11–13 LRRs and a N-terminal cysteine cluster sequence of C_X3_C_X_C_X6-17_C. Class IV SLRPs do not have ear repeats. CHAD harbors a N-linked keratan sulfate side chain, and clusters of lysine and arginine at its C-terminus. Molecular weight represents SLRP core proteins without glycosylation.

**Figure 2 ijms-22-07293-f002:**
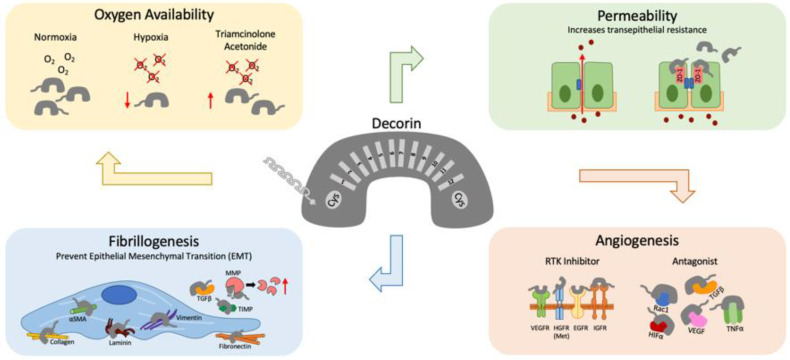
Decorin’s versatility in the retina. The class I horseshoe-shaped SLRP, decorin has a protein core made of 12 leucine rich repeats (LRRs) which are flanked by cysteine rich residues. Their GAG side chains are typically chondroitin/dermatan sulfate. Under hypoxic conditions, decorin levels decreased, which can be rescued with triamcinolone acetonide. Barrier permeability can be compromised during high glucose conditions. Decorin has been reported to increase transepithelial resistance, and increase tight junction proteins such as occludin and Zonula Occludens-1 (ZO-1). Decorin can also regulate angiogenesis by exhibiting both pro- and anti-angiogenic capabilities. Decorin is a pan-receptor tyrosine kinase (RTK) inhibitor and has been shown to bind and inhibit Rac1, HIFα, VEGF, TGFβ, and TNFα. Decorin can also reduce proteins involved in fibrillogenesis such as collagen, αSMA, laminin, vimentin and fibronectin. Furthermore, decorin activate matrix metalloproteinases (MMPs) and reduce tissue inhibitors of metalloproteinases (TIMPs) to help prevent epithelial mesenchymal transitioning (EMT).

**Table 1 ijms-22-07293-t001:** Distribution of SLRPs in the retina.

	**Biglycan**	Decorin	Fibromodulin	Lumican	PRELP/ Prolargin	Opticin	Osteoglycin/ Mimecan	Chondro-adherin (CHAD)	Tsukushi	Nyctalopin
**ILM**	Human [19]	Human [19]	Human [19]	Human [19]	Human [19]	Human [19]	Human [19]			
**NFL**	Human [19], Mice [20]	Human [19], Mice [20], Rat [21]	Human [19], Mice [20]	Human [19]	Human [19]	Human [19]	Human [19]			
**GCL**	Human [19], Mice [20]	Human [19], Mice [20], Rat [21]	Human [19], Mice [20]	Human [19], Mice [22]	Human [19]	Human [19]	Human [19]			Human [23], Mice [24], Rat [24], Chick [25]
**IPL**	Human [19], Mice [20]	Human [19], Mice [20], Rat [21]	Human [19], Mice [20]	Human [19], Mice [22]	Human [19]	Human [19]	Human [19]	Mice [26]		Mice [24], Rat [24], Primate [27], Rabbit [27]
**INL**	Human [19], Mice [20]	Human [19], Mice [20], Rat [21]	Human [19], Mice [20]	Human [19], Mice [22]	Human [19]	Human [19]	Human [19]		Mice [28]	Human [23], Mice [24], Rat [24], Chick [25]
**OPL**	Human [19], Mice [20]	Human [19], Mice [20], Rat [21]	Human [19], Mice [20]	Human [19], Mice [22]	Human [19]	Human [19]	Human [19]	Mice [26]		Mice [24], Rat [24], Primate [27], Rabbit [27]
**ONL**	Human [19]	Human [19], Mice [20], Rat [21]	Human [19], Mice [20]	Human [19]	Human [19], Mice [29]	Human [19]	Human [19]			
**PRL**	Human [19]	Human [19], Mice [20], Rat [21]	Human [19], Mice [20]	Human [19]	Human [19]	Human [19]	Human [19]	Mice [26]		Human [23]
**IPM**				Rat [30]						
**RPE**		Human [19], Rat [21]	Human [19]		Mice [29]	Human [19], Canine [31]				

ILM inner limiting membrane; NFL nerve fiber layer; GCL ganglion cell layer; IPL inner plexiform layer; INL inner nuclear layer; OPL outer plexiform layer; ONL outer nuclear layer; PRL photoreceptor layer; IPM interphotoreceptor matrix; RPE retinal pigmented epithelium.

## Data Availability

Data is available upon request.

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
