# Peer review of "Small Leucine-Rich Proteoglycans (SLRPs) in the Retina"

_ijms, 2021, doi:10.3390/ijms22147293_

Round 1

Reviewer 1 Report

This article addresses the importance of the SLRPs in the retina. The authors have reviewed the most important SLRPs described to be expressed in the retina and provided some evidences of expression of some new SLRPs not described so far. This review is interesting considering the implications of these proteoglycans in health and disease.

I think this review add important literature in the field. Overall the article is well written and well-illustrated.

Some minor points should be addressed:

- The authors mention that some SLRPs have important roles as pro and anti-angiogenic factors. Specifically, the authors described that biglycan is associate4d with increased scratch wound migration indicating a role in angiogenesis. More recently, it was described that biglycan is a a key player in regulating angiogenesis (number of vessel) using an in vivo model (CAM assay), as well as to regulate/correlate with the expression of several key angiogenic factors in human samples (as VEGFB, FGF2, ANGPT1, VEGFC, ANGPT2)(https://doi.org/10.3390/cancers13061330 ). Besides the reference 30 (Kinsella et al, 1997), this work should be mentioned as a more recent citation.

- In figure 1 and in the text whenever the molecular weight of the SLRPs are mentioned. The molecular weight that the authors described in the MW regarding the peptide moiety, without any glycosylation. For instance, biglycan have a predicted MW of 42 kDa; however due to the presence of GAGs its MW increased up to 150 (or more). The same happens with the other proteoglycans that are glycosylated. This important subject should be clarified by the authors or provide a note with the MW of these proteoglycans non glycosylated and fully glycosylated.

Figure 3. The authors described the mRNA levels of SLRPs in the retina. Some of them never described. Although I understand why the authors have provided this information in this review, I believe that is not appropriated for a Review article. The mRNA levels are not always related with the expression of the protein. A WB analysis will provide more solid information about the expression of all SLRPs in the retina. This data shoud be explored in the future and I will suggest remove the figure from this Review article.

Author Response

This article addresses the importance of SERPs in the retina. The authors have reviewed the most important SLRPs described to be expressed in the retina and provided some evidence of expression of some new SLRPs not described so far. This review is interesting considering the implications of these proteoglycans in health and disease.

I think this review add important literature in the field. Overall the article is well written and well-illustrated.

Thank you for your comments.

Some minor points should be addressed:

- The authors mention that some SLRPs have important roles as pro- and anti-angiogenic factors. Specifically, the authors described that biglycan is associate4d with increased scratch wound migration indicating a role in angiogenesis. More recently, it was described that biglycan is a key player in regulating angiogenesis (number of vessels) using an in vivo model (CAM assay), as well as to regulate/correlate with the expression of several key angiogenic factors in human samples (as VEGFB, FGF2, ANGPT1, VEGFC, ANGPT2)(https://doi.org/10.3390/cancers13061330 ). Besides reference 30 (Kinsella et al, 1997), this work should be mentioned as a more recent citation.

As suggested, we have made changes in the text (page 5, line 40) and added the reference.

- In figure 1 and in the text whenever the molecular weight of the SLRPs are mentioned. The molecular weight that the authors described in the MW regarding the peptide moiety, without any glycosylation. For instance, biglycan has a predicted MW of 42 kDa; however, due to the presence of GAGs its MW increased up to 150 (or more). The same happens with the other proteoglycans that are glycosylated. This important subject should be clarified by the authors or provide a note with the MW of these proteoglycans non-glycosylated and fully glycosylated.

Thanks for pointing it out. We have made changes in the text and figure legend.

Figure 3. The authors described the mRNA levels of SLRPs in the retina. Some of them never described. Although I understand why the authors have provided this information in this review, I believe that is not appropriated for a Review article. The mRNA levels are not always related to the expression of the protein. A WB analysis will provide more solid information about the expression of all SLRPs in the retina. This data should be explored in the future and I will suggest removing the figure from this Review article.

As suggested, we have removed figure 3.

Reviewer 2 Report

Interesting topic. There are too many repetitions and redundancies along the text, but the review could be globally considered valid. Just rewrite simpler the Abstract and, please, shorten the Introduction significantly. The aim of scientific articles is to involve readers in original or intriguing topic, without being redundant rephrase same concepts several times in different words.

Colored figures and long labels are useful to understand and sum up the role and function of SLRPs, which have been deeply investigated and classified in order to better explain their potential role about the triggering of some severe retinal diseases. Despite the fact this is a review, the reference list is maybe too long and could be shorten focusing on more relevant and updated references.

Author Response

Interesting topic. There are too many repetitions and redundancies along with the text, but the review could be globally considered valid. Just rewrite simpler the Abstract and, please, shorten the Introduction significantly. The aim of scientific articles is to involve readers in original or intriguing topic, without being redundant rephrase same concepts several times in different words.

Colored figures and long labels are useful to understand and sum up the role and function of SLRPs, which have been deeply investigated and classified in order to better explain their potential role about the triggering of some severe retinal diseases. Despite the fact this is a review, the reference list is maybe too long and could be shorten focusing on more relevant and updated references.

Thank you for your comments. The abstract has been edited as suggested and the introduction has been shortened with fewer references. 
